# Sensing Control Parameters of Flute from Microphone Sound Based on Machine Learning from Robotic Performer

**DOI:** 10.3390/s22052074

**Published:** 2022-03-07

**Authors:** Jin Kuroda, Gou Koutaki

**Affiliations:** Department of Computer Science and Electrical Engineering, Kumamoto University, Kumamoto 860-8555, Japan; kuroda@navi.cs.kumamoto-u.ac.jp

**Keywords:** parameter estimation, flute-playing robot, neural network, multilayer perceptron, learning to rank

## Abstract

When learning to play a musical instrument, it is important to improve the quality of self-practice. Many systems have been developed to assist practice. Some practice assistance systems use special sensors (pressure, flow, and motion sensors) to acquire the control parameters of the musical instrument, and provide specific guidance. However, it is difficult to acquire the control parameters of wind instruments (e.g., saxophone or flute) such as flow and angle between the player and the musical instrument, since it is not possible to place sensors into the mouth. In this paper, we propose a sensorless control parameter estimation system based on the recorded sound of a wind instrument using only machine learning. In the machine learning framework, many training samples that have both sound and correct labels are required. Therefore, we generated training samples using a robotic performer. This has two advantages: (1) it is easy to obtain many training samples with exhaustive control parameters, and (2) we can use the correct labels as the given control parameters of the robot. In addition to the samples generated by the robot, some human performance data were also used for training to construct an estimation model that enhanced the feature differences between robot and human performance. Finally, a flute control parameter estimation system was developed, and its estimation accuracy for eight novice flute players was evaluated using the Spearman’s rank correlation coefficient. The experimental results showed that the proposed system was able to estimate human control parameters with high accuracy.

## 1. Introduction

The most efficient way for a learner of a musical instrument to improve is to practice under the guidance of an instructor of that instrument. With advice from an instructor, the learner can more rapidly identify areas that need improvement and correct any problems. In addition, being able to listen to an example of correct performance makes it easier to have a clear idea of the desired musical outcome.

However, the time available for training is limited because of the need for face-to-face lessons and the fact that instructors take on multiple students. Therefore, learners need to improve their performance skills by practicing on their own. This is a difficult task for inexperienced students who have never learned a musical instrument before, and it can lead them to be hesitant to practice. For this reason, it is important to improve the quality of self-practice so that more people can be easily exposed to learning musical instruments, and many systems have been developed to assist self-practice.

A practice assistance system can not only measure and evaluate the tone of the performance, but also measure the physical state of the performer, including posture, fingering, and mouth shape, and provide feedback to the performer to assist in practice. We will refer to the posture and state of the instrument and performer as “control parameters”. A practice assistance system is a system that measures these control parameters and feeds them back to the performer.

Several practice assistance systems have been developed for keyboard instruments such as the piano [1], string instruments such as the violin [2], and wind instruments such as the saxophone [3]. Regarding the tone of the performance, some systems evaluate the timbre and stability of the performance [4], while others provide visual feedback of the timbre [3]. For control parameters other than sound, there is a system for string instruments that uses sensors to measure and feed back physical parameters such as the bowing motion of the performer [5].

The measurement and estimation of physical control parameters is possible for string and keyboard instruments, but difficult for wind instruments. This is because the physical control parameters of wind instruments include the position and shape of the tongue and lips, and the flow rate and angle of the breath, which cannot be externally observed. It is also impractical to place a sensor in the oral cavity to measure the control parameters, and attaching a sensor to the instrument would interfere with the natural playing of the performer.

In this study, we propose a method for estimating the control parameters of wind instruments, which is intended for application to wind instrument performance assistance systems. The proposed method can estimate the control parameters of a wind instrument from the sound of the human performance of a musical instrument. In addition, a deep learning method is used to estimate the parameters.

### 1.1. Basic Ideas

It is difficult to estimate control parameters from sound only; therefore, machine learning is often used to estimate them. However, as mentioned above, it is difficult to measure the control parameters of a wind instrument; therefore, it has not been possible to prepare training data for machine learning. To solve this problem, we used a robot that imitates the human performance of a wind instrument to obtain a large amount of training data consisting of performance recordings matched with control parameters. This made it possible to apply machine learning methods. In addition, because the instrument is played by a robot, it is possible to obtain accurate control parameters, such as breath flow and angle.

Based on this idea, we have developed a flute control parameter estimation system. Specifically, the system estimates the flow rate of the player’s breath and the angle of the breath hitting the instrument from the playing sound alone, and provides feedback to the player. This system is the first of its kind to estimate the control parameters of a wind instrument from sound alone.

### 1.2. Technical Overview

We developed the following technical elements in this system.

(a)Construct dataset of robot performer

One of the advantages of using the performance of a musical instrument-playing robot as a dataset is that the control parameters can be obtained as numerical values. To play a musical instrument, the robot needs to control components such as motors and flow valves. Therefore, the control values of the components can be regarded as the control parameters necessary for the performance. This makes it possible to obtain the control parameters of a wind instrument without using sensors, which has been difficult. In addition, the parameters can be gradually fine-tuned in the robot’s performance, and it is possible to obtain a large amount of data covering a variety of parameters.

(b)Combined robot and human performance training dataset

The instrument-playing robot developed in this study is a simple robot with the aim of changing its control parameters and producing sounds in a consistent manner, without consideration of whether or not its tones and expressions are rich. If there is a difference in timbre between the robot and human performances, training the model using only the robot performance may overadapt, and fail to estimate the human performance well. To solve this problem, we used a small amount of human performance as an additional training source to improve the adaptability to human performance.

(c)Using learning to rank for human training dataset

When human performance is used for model training, it is difficult to obtain the parameters as numerical values. However, for two performances with widely separated parameters, it is possible to determine the relationship between the parameters. Therefore, we adopted a ranking method to train the model so that it could correctly estimate the relationship between the parameters of two human performances.

## 2. Related Works

Many systems have been developed to assist in the practice of musical instruments, but the target instruments, methods of assistance, and equipment used are all different. In this section, we introduce the systems that have been developed with and without the use of control parameters.

### 2.1. A Practice Assistance Method Using Control Parameters

A study by David and Rafael was carried out to assist violin practice by using control parameters [6]. In this study, the bowing motion of a professional violin player was captured using motion capture to estimate the playing technique. In this study, the estimated results are fed back to the learner so that they can learn more appropriate playing techniques. Additionally, Miyazato et al. developed a system that provided more specific instructions [5]. In this study, the bowing motions of a learner and an advanced violin player were acquired and compared. Based on the feedback, the learner could improve their playing more specifically by approaching the movements of the advanced player.

Some studies have obtained control parameters and practical assistance with the use of wearable devices. In the study by Di Tocco et al., wrist and elbow movements were captured using sensors embedded in the garment [7]. From the acquired values, the acquired data allowed to study the timing of the bowing movements, which is especially useful for beginners. In a study by Van der Linden et al., haptic feedback using motor vibrations was implemented simultaneously with the acquisition of body movements [8]. Vibration feedback allows the learner to determine whether the performer is in the proper posture and bowing position. This study shows that vibration feedback is effective in improving the bowing techniques of beginners.

### 2.2. A Practice Assistance Method without Control Parameters

Practice assistance using control parameters is used for stringed instruments, percussion instruments, and keyboard instruments, where it is easy to obtain physical control parameters. For wind instruments, however, it is difficult to measure the breath or the inside of the mouth with a sensor because of the nature of the instrument, which is played by blowing. For this reason, most of the proposed practice assistance methods for wind instruments are based on feedback from performance recordings.

Romani et al. developed a system for evaluating the timbre of a performance [4]. In this work, a professional evaluated the timbre of various wind instrument performances and built a model that could provide the same evaluation as a professional. Additionally, Kimura et al. proposed a method to provide visual feedback of the timbre [3]. In this method, differences in saxophone pronunciation were extracted by a variational autoencoder, and visual feedback was provided to the learner. By using these methods, learners can evaluate their own performances, which can be used to improve their playing.

### 2.3. Flute-Playing Robot

Flute-playing robots have been developed in the past, the most representative of which was developed by Takanishi’s group [9,10,11]. This research aims to develop a humanoid performance robot capable of performing at a high level. The lips, oral cavity, and other parts of the robot are designed based on biology and have many degrees of freedom. This robot is also intended to interact with humans through music. Therefore, it is equipped with an auditory feedback system and other features that allow the performing robot and human to play.

In contrast, the purpose of our study is to use robot performances for machine learning to estimate the parameters of human performances. To accomplish this, we developed a flute-playing robot with the minimum parameters necessary for performance. Therefore, the robot developed by us and that developed by Takanishi et al. are clearly different on research subjects. Our method will be used for parameter expansion based on the number of parameters of the robot, and high-performance robots such as those developed by Takanishi et al.can be used with our method. 

### 2.4. Position of This Study

Systems that provide practice assistance, distinguished by their characteristics, are shown in Figure 1. Each system is broadly classified according to whether or not it measures control parameters. They can be further classified into systems that can provide control parameters or systems that only provide evaluation and estimation. In the case of stringed instruments, where physical control parameters can be obtained, it is possible to construct a system that provides specific teaching and evaluation. However, for wind instruments, where it is difficult to obtain physical control parameters, there are many systems that provide feedback and evaluation, but none that can provide teaching of control parameters.

In this study, we propose a new control parameter estimation system that can provide guidance on the physical parameters without measuring the control parameters.

## 3. Control Parameter Estimation System

In this section, we describe the proposed control parameter estimation system and the dataset used for training.

### 3.1. Overview of the Proposed System

In this study, we developed a control parameter estimation system for the flute. The flute is classified as a woodwind instrument. When playing the flute, sound is produced by applying the breath to the edge of the embouchure hole. In this study, the flow rate and angle of the breath to the edge were defined as the control parameters. Therefore, the proposed system estimates these two parameters during the performance. The input of the system is the sound of the instrument playing, and the output is the breath flow rate and angle of the breath. Here, the system is limited to performance practice using only the flute head joint. This is a simple practice without fingering for beginners who have just started learning the flute.

### 3.2. Elements of the System

The overall diagram of the control parameter estimation system is shown in Figure 2. The system consists of dataset creation, model training, and control parameter estimation. In this section, each element is explained.

(a)Dataset creation

First, we created a training dataset based on the performances of a musical instrument-playing robot and a human. Two types of datasets were used for training. One was the robot performance dataset created by the flute-playing robot. This dataset contained the performance recordings and the performance control values of the robot. The performance control values of the robot were considered to be control parameters.

The other dataset contained human performances. The human performance dataset contained the recordings of the performances and labels of the corresponding control parameters. The labels did not have any numerical values, and the dataset was small in comparison to the robot performance dataset.

(b)Model training

In the model training, we used two datasets to determine the relationship between the performance and control parameters. The input to the model was a log-mel spectrogram generated from the audio recordings of each dataset. In the case of learning using the robot performance dataset, the model was trained by regression estimation using the control parameters as target values. On the other hand, in learning with the human performance data, a ranking method was used to learn the relationship between the control parameters of the two datasets. To perform these two types of learning simultaneously, we proposed a new loss function, which is the mean square error plus the loss function of the ranking method.

We adopted a learning model based on the MLP-Mixer [12]. The MLP-Mixer is a simple model that uses only multilayer perceptron (MLP) and does not use convolutional neural networks (CNNs), which are primarily used in conventional tasks using images.

(c)Parameter estimation

After the model training was completed, the trained model was used to estimate the control parameters. When a human performance recording was input to the trained model, the estimation results of each control parameter were output. This made it possible to obtain the human control parameters from the performance recording.

### 3.3. Flute-Playing Robot

We created a new flute-playing robot to collect recordings of its performance for use in learning. Two types of robots exist that can play musical instruments: those that aim to generate performances without reproducing human physical characteristics [13,14,15], and those that aim to achieve a more human-like performance by reproducing human body parts more precisely [16,17]. The latter type of robot can reproduce tones and playing techniques closer to those of humans, but it is more costly to develop. In this study, because the target is a single note performance with only the head joint, there is no need for advanced performance techniques or expression. For this reason, we developed a robot with a simple performance function that can play a single note using only the flute head joint.

An overall diagram of the flute-playing robot is shown in Figure 3. The motor, pump, and other parts used in the robot are shown in Table 1. This flute-playing robot consists of an angle-control mechanism to which the flute body is attached, a flow-control mechanism to send air to the flute, and a control circuit to control the robot. The robot is controlled by serial communication, and the flow rate and angle of the breath can be changed in real time.

#### 3.3.1. Angle-Control Mechanism

The angle-control mechanism includes a holder that keeps the flute head joint in place and can be rotated to adjust the angle of the air being blown into the head joint. In addition, there is a blowhole to send air from the air circuit to the flute head joint. The angle-control mechanism was designed using 3D CAD software. Subsequently, we created the necessary parts based on the designed model and assembled them. The base and the parts to hold the rotation motor were made from cut-out acrylic sheets. The other parts were made using a 3D printer, from PLA material.

The flute head joint was placed in the center of the angle-control mechanism, and two props were attached to support the head joint. The two props were mounted on two rails on the base and could slide freely in the horizontal direction. This allowed the head joint to be held in an optimal position. The flute was also equipped with a connector to attach to a servo motor. By rotating the head joint with the motor, the robot could blow air at various angles. The servo motor to rotate the flute and the parts to mount the motor to the base were placed in a straight line with the props supporting the flute. The motor was also mounted on a rail on the pedestal so it could be moved horizontally, and also allowed vertical height adjustment.

This allowed the motor to be correctly aligned with the connector attached to the flute head joint.

The blowhole was made in the shape of a square pyramid. The cavity inside was designed to become narrower towards the outlet, similar to the human mouth. At the back of the blowhole was a fitting that connected to the flow-control mechanism. A rubber sheet with an oval hole was attached to the front side of the blowhole. When a person plays a flute, the lips are deformed to the proper shape to regulate the flow of breath. The oval hole in the rubber sheet reproduces this feature.

The blowhole was attached to a horizontal rail to which the flute props and motor were mounted. The rail and blowhole were connected by an arm with two levels of articulation. This made it possible to adjust the position of the blowhole and flute three-dimensionally to achieve a stable sound.

#### 3.3.2. Flow-Control Mechanism

When playing a musical instrument, it is necessary to blow air at an appropriate flow rate to produce a stable sound. Therefore, it is also necessary for a musical instrument-playing robot to precisely control the airflow. To achieve this, we aimed to create a flow-control mechanism that could control the flow rate using a proportional flow valve.

The flow-control mechanism we created consisted of a pump for pumping air, a proportional flow valve to control the flow rate, and an air tank to rectify the air. The tubes connecting each component were standardized with an inner diameter of 4 mm. The proportional flow valve adjusted the airflow from the pump to an appropriate level. The proportional flow valve generated vibrations during its operation, and these vibrations were transmitted through the air pumped into the instrument. To remove the vibrations, an air tank for rectification was added between the proportional flow valve and the instrument. The air tank was made from a 500 mL plastic bottle. Two tubing fittings were attached to the air tank. The temperature and the humidity of the robot’s airflow were approximately 18 °C and 30%, respectively.

#### 3.3.3. Control Circuit

The control circuit had a microcontroller for control and a driver circuit for the flow valve. The servomotor was controlled by a PWM signal output from the microcomputer. However, the proportional flow valve was controlled by the current value, but the microcomputer we used could not output the current value in analog form. Therefore, a driver for the proportional flow valve was used to convert the voltage value into a current value for flow control.

### 3.4. Robot Dataset

In this section, we describe the dataset created using the flute-playing robot.

#### 3.4.1. Recording Environment

The performances by the flute-playing robot were recorded in a soundproof room. Only the robot and the recording equipment were installed in the soundproof room, and the recording was carried out by remote control. A microphone attached to the robot was used for recording.

#### 3.4.2. Control Parameters

Two control parameters were required to be estimated in this system: the flow rate of the breath during playing and the angle. Therefore, we indirectly extracted the control parameters from the flute-playing robot by considering each control value as a control parameter. To obtain performances with various combinations of control parameters, we created a grid of data by making stepwise changes to each parameter in turn. The upper and lower limits of the parameters were the limits at which sound could not be created properly. The parameters were changed during the performance, resulting in 26 levels of breath flow and 15 levels of breath angle. This resulted in a total of 390 combinations of parameter values.

#### 3.4.3. Number of Data

Each parameter combination was recorded for 0.5 s of performance, and 100 data values were recorded for each combination of parameters. Therefore, 39,000 data points were recorded, and the total recording time was 325 min.

### 3.5. Human Performance Fixture

When a human plays the flute with the instrument held in their hands, the angle and position of the flute changes each time they played it. This causes the angle of the breath to change, making it difficult to acquire an accurately labeled performance. To solve this problem, we developed a device to fix the flute to the human head with the aim of reducing the change in the angle of the breath.

The fixation device is shown in Figure 4. The apparatus is divided into a flute fixation stand and a head fixation stand. The former is placed on the floor and the latter on a desk. Each device was made by attaching an acrylic plate and parts made with a 3D printer to a stand for smartphones. The flute head was attached to the flute-fixing stand with rubber bands, and the angle of the flute could be adjusted by rotating it. The stand for head fixation had two parts: one for placing the chin and the other for placing the forehead, which enabled the head to be held in a fixed position.

### 3.6. Human Dataset

Using the fixation device, a human performance dataset was created. The performances were carried out by one of the authors, plural. The recording location and microphones used were the same as those used in the recording of the flute-playing robot. Two parameters were used for the performance: breath flow rate and breath angle.

#### 3.6.1. Parameter Labels

In the dataset from the flute-playing robot, we created data with multiple levels of breath flow rate and breath angle by changing the control values. However, when recording human performance, it is difficult to accurately distinguish between multiple levels of each parameter. Therefore, in the human performance data, each parameter was labeled with two levels that could be clearly distinguished. For breath flow rate, the label used for a high breath flow rate is “F” and that for a low flow rate is “P”.

For the angle of the breath, each label and the corresponding angle are shown in Figure 5. As shown in Figure 5a, when the flute is rotated toward the back, as seen by the player, the lip plate of the flute approaches parallel to the direction of the breath. This angle was labeled “BACK”. Conversely, in Figure 5b, the flute was rotated toward the front, and the lip plate of the flute was closer to perpendicular to the direction of the breath. This angle was labeled “FRONT”. The difference between the labeled angles in the human performance data was approximately 10°, which is approximately the same as the maximum difference in rotation angles in the performance data of the flute-playing robot.

Based on the above, we recorded four sets of data by combining the values of each parameter.

#### 3.6.2. Number of Data

Each data sample was recorded for 0.5 s, and 80 data samples were created for each combination of parameters. Therefore, the total number of recorded data samples was 320, and the total recording time was 160 s.

### 3.7. Dataset Creation Procedure

From the recorded robot and human performances, we created a mel spectrogram image as an input into the model, which was used as the training dataset. The procedure for creating the training dataset is shown in Figure 6. The Python library, librosa [18], was used for data preparation. The procedure was as follows:From the recorded performance, 0.5 s of performance was cut out.The cut-out recorded performance was then converted into a mel spectrogram image.Finally, the power spectrum of the mel spectrogram image was converted to decibels.

Thus, the image output is a 128 × 128 square. As shown in Figure 6, the horizontal axis of the output image represents the time, and the vertical axis represents the frequency. The pixel value of the image indicates the strength of the frequency component of the performance. The number of images in the created dataset was 39,000 for the robot performance data and 320 for the human performance data.

### 3.8. Control Parameter Estimation Model

In this section, we describe the model structure of the control parameter estimation model and the loss function used for training.

#### 3.8.1. Model Structure

Figure 7 shows the structure of the flute control parameter estimation model. The model estimates a single control parameter from a log-mel spectrogram image generated from a recording. The estimation accuracy of each parameter can be improved by preparing multiple models for each parameter to be estimated.

We used the MLP-Mixer [12] as the base of the model. The MLP-Mixer divides the input into patches and alternately performs feature extraction within and between patches. The MLP layer consists of two MLP blocks, and feature extraction is performed on the MLP blocks. The MLP-Mixer is more computationally efficient than existing models such as CNNs and Attention, and can achieve comparable performance for training tasks with a sufficient amount of training data. In this study, we constructed a learning model with three MLP layers. The MLP-Mixer was implemented using the PyTorch library (https://github.com/lucidrains/mlp-mixer-pytorch, accessed on 23 June 2021).

The output from the model was a value between 0 and 1, corresponding to the magnitude of the estimated parameters. For the breath flow rate, the closer to 0, the lower the flow rate, and the closer to 1, the higher the flow rate. In contrast, the angle of the breath approached 0 as it rotated toward the label “BACK”, and approached 1 as it rotated toward the label “FRONT”.

#### 3.8.2. Loss Function

The control parameter estimation model estimated one parameter and learned the robot and human performance data simultaneously.

In the robot performance dataset, there were values of control parameters, such as flow rate and angle, which were identified as correct answers. Therefore, model training by regression estimation was possible. For the loss function of the robot performance data, we adopted the mean squared error (MSE), which is a loss function often used in regression estimation. The MSE is expressed by the following equation: (1)$$\mathrm{MSE}=\frac{1}{n}\sum_{i=1}^{n}{\left(y_{i}-{\hat{y}}_{i}\right)}^{2}$$
where *n* is the number of data samples in the batch, $y_{i}$ is the value of the control parameter that is the correct answer, and ${\hat{y}}_{i}$ is the control parameter estimate output by the model.

On the other hand, the control parameters of human performance data are labels, and although the labels indicate large and small values of the control parameters, they do not have exact numerical values. Therefore, we introduced the loss function of RankNet [19], a ranking method that learns the relationship between two datasets.

RankNet learns the relative ordering relationships for pairs of data. Given two pairs of human performance data $x_{1},x_{2}$, we denoted $x_{1}▹x_{2}$ when the output parameter estimation result from model f(x) was $f\left(x_{1}\right)>f\left(x_{2}\right)$. When human performance data $x_{i}$ and $x_{j}$ are input, the difference in the output control parameter values is denoted as $O_{ij}$, and is defined as follows: (2)$$O_{ij}=f\left(x_{i}\right)-f\left(x_{j}\right)$$

In this case, the probability $P_{ij}$ such that $x_{i}▹x_{j}$ is expressed as follows: (3)$$P_{ij}=P\left(x_{i}▹x_{j}\right)=\frac{1}{1+\exp\left(-O_{ij}\right)}$$

RankNet is trained such that this probability $P_{ij}$ matches ${\bar{P}}_{i}j$, which represents the relationship between the two datasets. Using cross entropy, the loss $C_{ij}$ for the performance data $x_{i}$ and $x_{j}$ can be expressed as follows: (4)$$\begin{array}{cc} C_{ij} & =-{\bar{P}}_{ij}\log P_{i}j-\left(1-{\bar{P}}_{ij}\right)\log\left(1-P_{ij}\right)\\ \; & =-{\bar{P}}_{ij}O_{ij}+\mathrm{softplus}\left(O_{ij}\right) \end{array}$$

In this case, ${\bar{P}}_{ij}$ is as follows
(5)$${\bar{P}}_{ij}=\left\{\begin{array}{cc} 1 & \left(x_{i}▹x_{j}\right)\\ 0 & \left(x_{j}▹x_{i}\right)\\ \frac{1}{2} & \left(\mathrm{otherwise}\right) \end{array}\right.$$

The labels for the control parameters in the human performance data were “F” and “P” for flow rate and “FRONT” and “BACK” for angle, with the former labeled large and the latter labeled small.

Based on the above, when the *i*th pair of human performance data in a batch is $a_{i},b_{i}$, the loss function of this study can be expressed as follows: (6)$$\mathrm{loss}=\frac{1}{n}\sum_{i=1}^{n}{\left(y_{i}-{\hat{y}}_{i}\right)}^{2}+\alpha \left(-{\bar{P}}_{a_{i}b_{i}}O_{a_{i}b_{i}}+\mathrm{softplus}\left(O_{a_{i}b_{i}}\right)\right)$$
where *n* is the batch size, *y* is the output when the robot data is input, $\hat{y}$ is the control parameter of the robot, $O_{ab}$ is the difference in the output when a pair of human data is input, and ${\bar{P}}_{ab}$ is the relationship between the parameters of the two human data, large and small.

The weight of each loss function can be changed by varying the value of the coefficient α. Because the human performance dataset was small compared to the robot performance dataset, an excessively large value of α could cause overtraining. For this reason, we set α=0.1 to prevent overtraining using human performance data.

## 4. Evaluation Experiment

In this section, to evaluate whether the proposed model could correctly estimate human performance parameters, we conducted performance parameter estimation experiments with several participants.

### 4.1. Experimental Conditions

#### 4.1.1. Experimental Models and Training Conditions

To verify the effectiveness of the proposed method, we compared it with several baseline models. For the baseline models, we used the CNN-based models AlexNet [20], MobileNetV2 [21], and ResNet18 [22]. In addition, to compare the estimation results of the models due to differences in the datasets used, we trained the models using only one dataset, either the robot performance dataset or the human performance dataset, and using both of them. The training conditions for each model were the same, with the number of training epochs set to 100 and the batch size set to 16. We used Adam as the optimization function and set its learning rate to 0.00005.

#### 4.1.2. Performance by Several Participants

To evaluate the performance of the model, we asked several participants to play the flute to create an experimental dataset. There were eight participants (p1 to p8), all of whom were beginners in flute playing. The eight participants were adult males between the ages of 21 and 24. Each participant played with several levels of breath flow and angle. Breath flow rate was labeled either “F” for playing with a high flow rate and “P” for playing with a low flow rate. The angle of the breath was labeled at three levels, with “CENTER” as the standard for playing at the most comfortable angle, “FRONT” for rotating the flute to the front, and “BACK” for rotating the flute to the back. To evaluate the model for each parameter, performances with varying angles and flow rates were recorded separately. The flute used in the experiment was a YAMAHA YFL-211, which was the same as the flute attached to the robot. Recordings were made using the following procedure:Recording was performed while changing the flow rate of the breath. The angle of the breath was set to the angle at which the participant could play the most naturally. The flow rate of the breath was changed in two steps, and approximately 10 samples were recorded for each step, for a total of approximately 20 samples.Recording was performed while changing the angle of the breath. The flow rate of the breath was set to what the participant could play most naturally. The angle of the breath was changed in three steps, and approximately 10 samples were recorded for each step, totaling approximately 30 samples.

For each parameter, the ideal plot results are shown in Figure 8. The expected output of the breath flow rate is such that “F” is plotted above “P”. On the other hand, the expected output of the breath angle is such that “BACK”, ”CENTER”, and “FRONT” are aligned from left to right.

#### 4.1.3. Evaluation Criteria

We used Spearman’s rank correlation coefficient [23] as a quantitative measure of the estimation results from each model. Spearman’s rank correlation coefficient is a criterion for correlation obtained from the two rank data. In this experiment, the rank obtained from the parameter labels assigned to the performance and the values estimated by the model were used. The angle parameter labels were assigned to “FRONT”, “CENTER”, and “BACK” in descending order of rank, and the flow parameter labels were assigned to “F” and “P” in descending order of rank. The parameter estimates output from the model were defined as higher values equating to higher ranks.

The correlation coefficient is expressed as a value between –1 and 1, with a value closer to 1 indicating a higher positive correlation. In the ideal plot, the correlation coefficients are 1. Therefore, the higher the value of the correlation coefficient, the more accurately the performance parameters are estimated.

### 4.2. Experimental Results

Examples of the flow parameter estimation results are shown in Figure 9, and the angle parameter estimation results are shown in Figure 10. The correlation coefficients for the performance parameter estimation results for each participant are presented in Table 2. Results for baseline models and the proposed model are shown. The dataset entries indicate the use of the robot and human performance datasets, with checkmarks indicating use and indicating non-use.

### 4.3. Discussion

#### 4.3.1. Result of Flow Rate Estimation

In the flow estimation, the correlation coefficient was highest when the MLP-Mixer was used as the model and trained only with the robot performance data. The estimation example with the highest correlation is shown in Figure 9b. In the estimation example, it can be confirmed that the flow rate parameters can be clearly separated. In all models, the correlation was the highest when only the robot performance dataset was used. This indicates that the flow parameters can be estimated by learning using the robot performance dataset, and that using the robot performance as training data was an effective method.

In contrast, the model trained using only human data did not provide very accurate estimates, as shown in Figure 9a. The method that used both robot and human performance data sometimes resulted in a narrower separation than the method that used only the robot performance dataset, as shown in Figure 9c. From these results, it can be concluded that the use of the human performance dataset was not very effective for flow estimation. The flow rate labels used to create the human performance dataset were assigned based on the subjectivity of the performer. Therefore, compared to the angle labels, the variability of the data was larger, and it is possible that the features could not be learned well.

#### 4.3.2. Result of Angle Estimation

In the angle estimation, the proposed method, which used the MLP-Mixer as a model and simultaneously used robot performance data and human performance data, showed the highest correlation. The estimation example with the highest correlation is shown in Figure 10c. It can be seen that each parameter can be clearly separated by label. In addition, the separation and relationship between the labels were accurately identified.

In contrast, the method using only the robot performance data for training, as shown in Figure 10a, did not achieve accurate separation. The method that used both robot and human performance data simultaneously, but used baseline models other than MLP-Mixer, was not able to correctly learn the relationship between the labels, although it was able to separate them to some extent, as shown in Figure 10b. Therefore, we can say that using both robot performance data and human performance data for training simultaneously and using the MLP-Mixer as the training model contributes to the improvement of the estimation performance.

#### 4.3.3. Limitation

An example of a failed estimation using the proposed method is shown in Figure 11. In the performance where the estimation failed, the labels “FRONT” and “CENTER” could not be separated well. This may be due to the large difference in tone quality between the human and robot performances of “FRONT” and “CENTER”.

In human flute playing, the closer the angle of the breath is to the “FRONT” direction, the larger the area of the lip plate that is blocked by the lips, and the tone changes. However, in the flute playing robot created in this study, there was no part corresponding to the human lips. Therefore, the timbre changes caused by the lips were not fully taken into account, and the discrepancy between the timbres of human and robot performances at the “FRONT” angle was large. This is considered to have prevented accurate estimation. To solve this problem, it is necessary to create a more humanlike robot with lips and other physical features.

In addition, in this study, only the author’s performance recording was used for the training data of human performance, and it is possible that there was not enough data to sufficiently represent the changes in characteristics when the angle of the breath was changed. Therefore, we believe that the performance can be improved by adding data from multiple human performances to the training.

#### 4.3.4. Future Work

(a)Reproduction of human breath

The exhaled air for a human playing the instrument has a special airflow with a temperature of approximately 37 °C and humidity of almost 100%. However, as shown in Section 3.3.2, the robot developed in this study breathed room temperature and dry humidity. Therefore it cannot reproduce the effect of the special airflow on the tone because it can only control the flow rate and angle of breath. To increase the quality of reproduction by the robot, it is necessary to improve the robot’s ability to control the temperature and humidity of air.

(b)Parameter expansion

The blowhole of the flute-playing robot created in this study has a very simple structure. The mouth of a human has complex parameters such as the shape and viscosity of the oral cavity and the shape of the lips. The accuracy of parameter estimation can be further improved by creating a sophisticated artificial mouth that takes these features into account, such as in [16].

The estimation system proposed in this study can be used to build as many models as the number of estimated parameters. Therefore, the framework in this study can be used to realize the extension of the estimated parameters using a robot that exhibits better performance.

(c)Application of this study

Our method can be applied to parameter-based practice-assist methods that have been developed for non-wind instruments. Although the estimation and investigation of the playing methods using parameters and feedback from the vibrations mentioned in related research were intended for stringed instruments, the application of our method makes it possible to use these methods for wind instruments as well. In addition, obtaining control parameters for wind instrument playing will enable a better understanding of the relationship between physical movements during playing and timbre.

## 5. Conclusions

In this study, we proposed a system for estimating physical control parameters from the sound of human performance, which is intended to be applied to practice assistance systems. The proposed system used the control values of a robot playing a musical instrument as a substitute for the control parameters, which enabled indirect acquisition of the control parameters from the sound of a recording without using special sensors such as a rotary encoder, flow, and pressure sensors. In addition, in order to prevent excessive adaptation of the model to the robot performance, we proposed the use of human performance data with clear parameter differences for training, and we introduced a ranking-based loss function.

To verify the effectiveness of the proposed system, we constructed a flute performance parameter estimation system for flute head joint sound recordings and evaluated its performance. As a result of estimating the physical control parameters of several participants, the proposed system was able to obtain high estimation accuracy and showed that the control parameters could be estimated from microphone sound recordings alone. These results show that it is possible to provide instruction on the performance parameters of wind instruments. In addition, we believe that our system can be applied not only to the acquisition of instrumental performance skills, but also to the acquisition of various other skills.

## Figures and Tables

**Figure 1 sensors-22-02074-f001:**
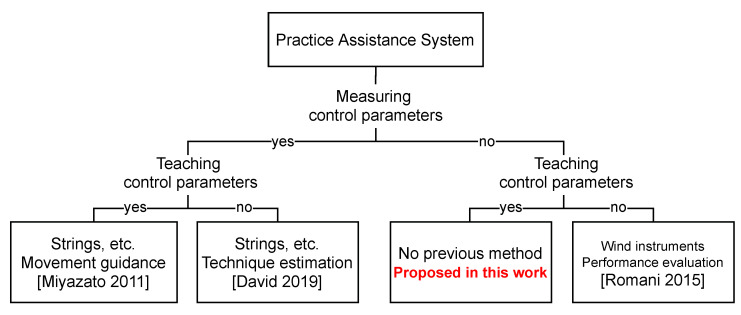
Classification of musical instrument practice assistance systems. In this study, we propose a new system that can provide movement teaching without the need to measure control parameters using sensors.

**Figure 2 sensors-22-02074-f002:**
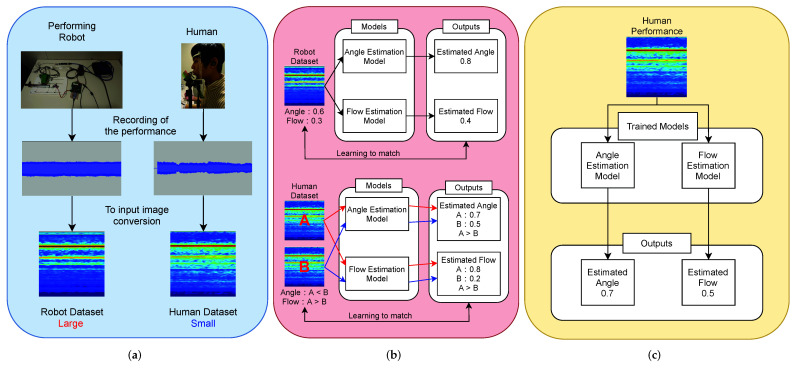
System diagram of the control parameter estimation system. Data were created from a flute playing robot and human performance. We used the dataset to train a model for control parameter estimation, which was separated for each estimated parameter. The trained models were used to estimate the parameters of human performance. Thus, it is possible to obtain control parameters only from the performance recordings. (**a**) Dataset Creation. (**b**) Model Training. (**c**) Parameter Estimation.

**Figure 3 sensors-22-02074-f003:**
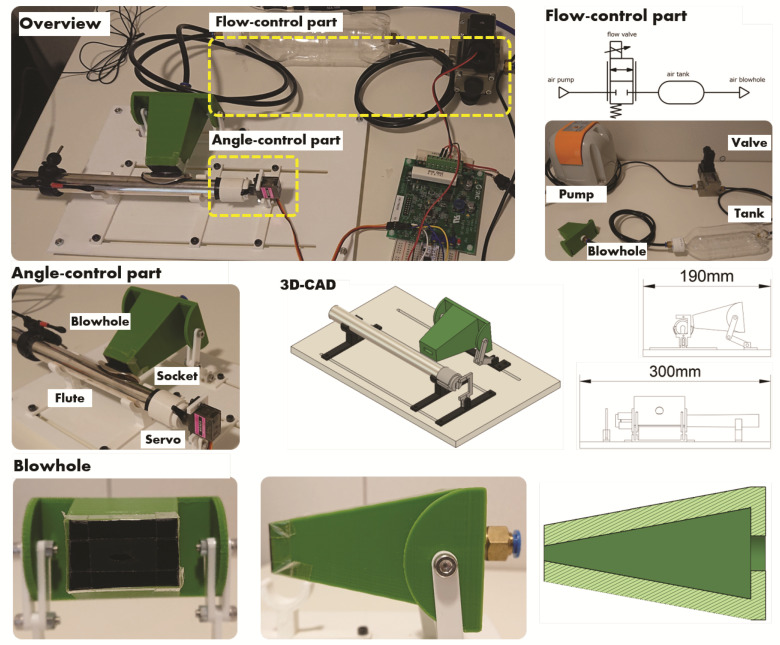
Overall diagram of the flute-playing robot. A servo motor is connected to the flute body, and the angle at which the breath is blown into the instrument can be adjusted by rotating the motor. In addition, the flow rate of the breath into the instrument can be adjusted by adjusting the opening of the flow valve in the air circuit.

**Figure 4 sensors-22-02074-f004:**
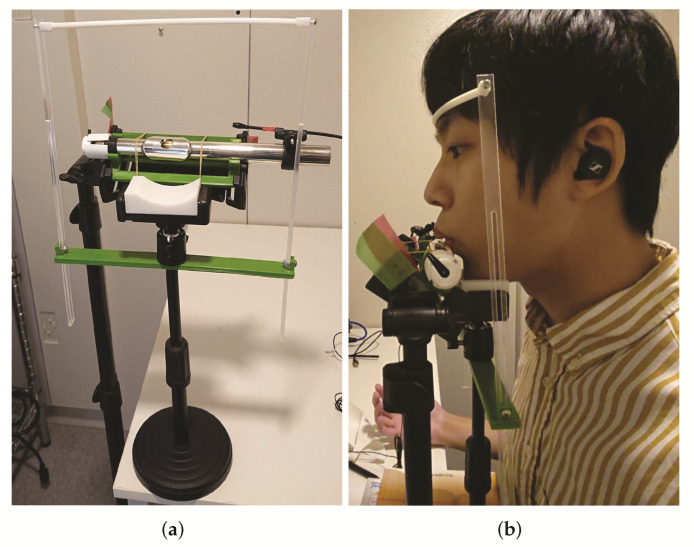
A head fixation device was developed for collecting human performance data. (**a**) Head fixer device. (**b**) Usage example.

**Figure 5 sensors-22-02074-f005:**
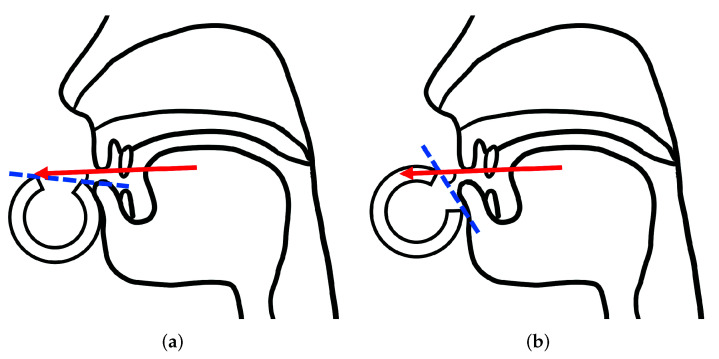
Angle labels for human performance. (**a**) Label “BACK”. (**b**) Label “FRONT”.

**Figure 6 sensors-22-02074-f006:**
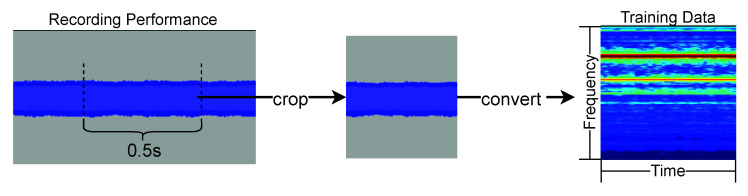
Data creation procedure.

**Figure 7 sensors-22-02074-f007:**
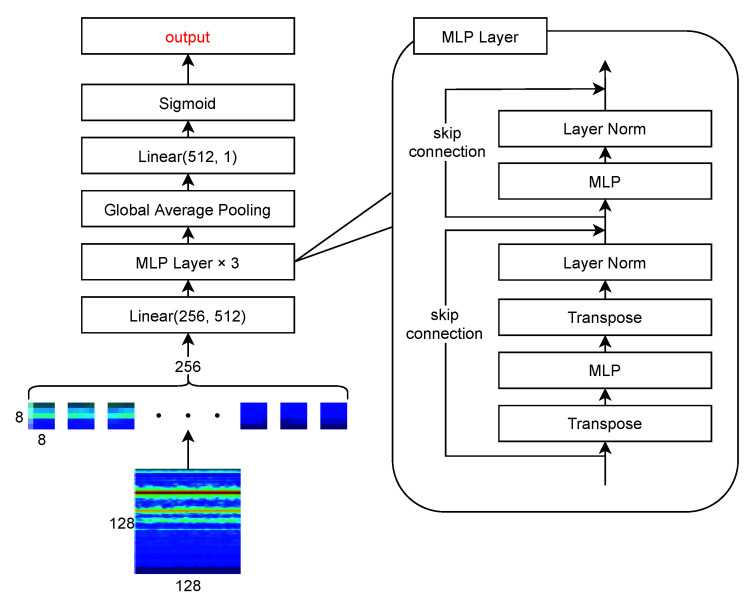
Structure of the parameter estimation model. We used the MLP-Mixer as the base of the model. The model estimates a single control parameter from a log-mel spectrogram image generated from a recording. The output from the model was a value between 0 and 1, corresponding to the magnitude of the estimated parameters.

**Figure 8 sensors-22-02074-f008:**
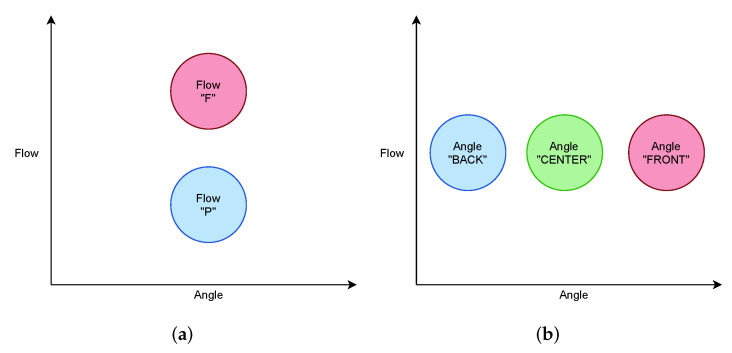
Ideal plot results for each parameter. (**a**) Ideal plot of flow rate. (**b**) Ideal plot of angles.

**Figure 9 sensors-22-02074-f009:**
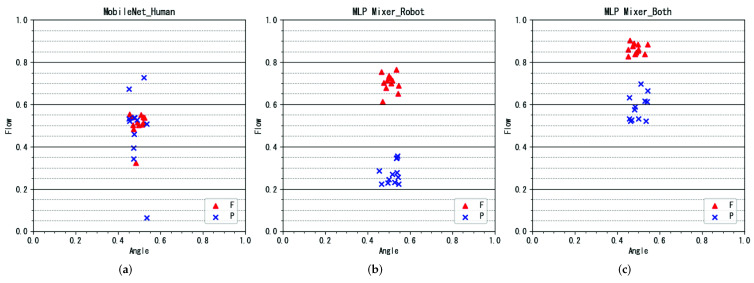
Examples of flow estimation results. (**a**) MobileNet and human dataset (conventional method). (**b**) MLP-Mixer and robot dataset (proposed method). (**c**) MLP-Mixer and robot + human datasets (proposed method).

**Figure 10 sensors-22-02074-f010:**
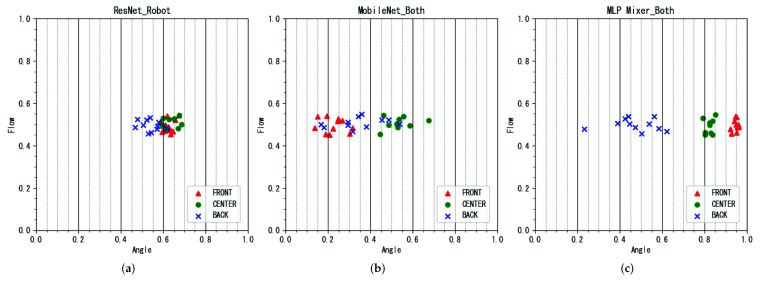
Example of angle estimation results. (**a**) MobileNet and human dataset (conventional method). (**b**) MLP-Mixer and robot dataset (proposed method). (**c**) MLP-Mixer and robot + human datasets (proposed method).

**Figure 11 sensors-22-02074-f011:**
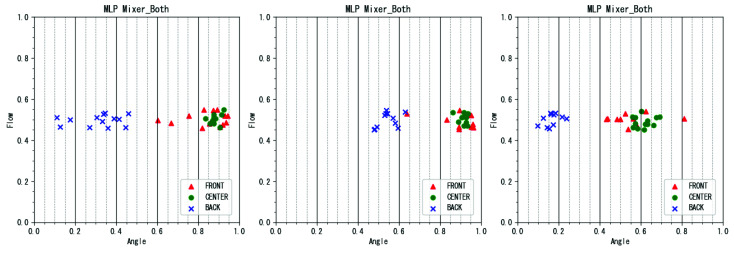
Example of failure of angle estimation by the proposed method.

**Table 1 sensors-22-02074-t001:** Parts used in the flute-playing robot.

Parts	Manufacturer	Model Number
Flute	YAMAHA	YFL-211
Flow valve	SMC	VEF2121-1-02
Flow valve driver	SMC	VEA250
Air pump	Yasunaga Air Pump	VDW20JA
Servomotor	Kyohritsu Electronic Industry	MG90S
Microcontroller	STMicroelectronics	NUCLEO-F303K8
Microphone	JTS	MA-500

**Table 2 sensors-22-02074-t002:** Spearman’s rank correlation coefficients for different models and datasets.

Model	Dataset		Correlation	
	Robot	Human	Flow	Angle
AlexNet (Baseline1)	√	–	0.320	0.225
	–	√	0.414	0.053
	√	√	0.021	0.636
MobileNet (Baseline2)	√	–	0.839	−0.008
	–	√	0.316	−0.187
	√	√	0.668	0.249
ResNet (Baseline3)	√	–	0.852	0.743
	–	√	0.602	0.305
	√	√	0.821	0.213
MLP-Mixer (Proposed)	√	–	**0.864 **	0.293
	–	√	0.340	0.784
	√	√	0.855	**0.809**

## Data Availability

The data and code used for training and experiments are available on Github (https://github.com/Jinx-git/Estimation-of-Control-Parameters-Using-a-Flute-Playing-Robot, accessed on 23 June 2021).

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
