# Peer review of "Sensing Control Parameters of Flute from Microphone Sound Based on Machine Learning from Robotic Performer"

_sensors, 2022, doi:10.3390/s22052074_

Round 1

Reviewer 1 Report

Dear authors, I read with much interest your original work presenting a robotic system for monitoring and training flute playing. I am convinced that readers will benefit from your work, which contributes significantly to an increasingly popular multidisciplinary line of research that combines musicology with bioengineering. While I think that the overall quality of the paper is acceptable, I believe that there are some major issues to fix before considering it for publication. Herebelow I list some hopefully useful comments/suggestions: 

Major points: 

1) I assume that two additional parameters impact on sound production. I am thinking about temperature and humidity of the air flow, which are quite peculiar in humans (37 degrees and almost 100% humidity). I would thus recommend to control these parameters in the air emitted by the robotic player. And I would also add this point in the final discussion.

2) To me, the shape of the plastic mouth seems an oversimplification of the complex 3d mouth cavity. I think also lubrication and viscosity influence air flow in the mouth (see point 1 above). Couldn’t the anatomy of the mouth be taken into account when creating the “artificial mouth”? I also think that such shape can dynamically change due to tongue movements. Finally, consider that humans might assume different lip posture or flute inclination to compensate for alteration in the air flow in the inner cavities.

3) Can you provide an additional figure of the blowhole? (e.g., front view)

4) I recommend to enrich the discussion comparing your findings with similar researches and, more in general, opening to the context of music technology.

5) Suggested refs:

Di Tocco, J., Massaroni, C., Di Stefano, N., Formica, D., & Schena, E. (2019). Wearable system based on piezoresistive sensors for monitoring bowing technique in musicians. In 2019 IEEE SENSORS (pp. 1-4). IEEE.

Van Der Linden, J., Schoonderwaldt, E., Bird, J., & Johnson, R. (2010). Musicjacket—combining motion capture and vibrotactile feedback to teach violin bowing. IEEE Transactions on Instrumentation and Measurement60(1), 104-113.

Minor points: 

Abstract: I would at least mention the fact that finally you tested your system with 8 human participants.

Line 23: “quickly”, I would say “more rapidly”

Line 24: “to understand what a good sound is”. I suggest: “to have a clear idea of the desired musical outcome”

Line 26: “instruction”, may be "training"?

Line 81: “which has been difficult in the past.” Can you support this with a ref?

Line 119: “Romani it et al.developed” typo here?

Line 120: “Kimura et al. . proposed” typo here?

Line 129: “provide teaching” I would say: “can provide”

Lines 138-144: Is basically a repetition of info the reader has clearly in mind at that point of the paper (there are other repetitions here and there, you might consider to shorten the text during revision).

Section 3.4.3. How could the 39000 data points result in 3 mins? (compare to 3.6.2. in which data were 320 and resulted in 160 s…) Must be wrong.

Line 302: By “one” of the author?

Line 400: additional information about participants would be probably appreciated (M/F, age..)

Reviewer 2 Report

In this paper, a machine learning method is used to develop a practice assistance system for learners. The input is the recording of the flute, and the output is the angle and the flow rate of the breath. When acquiring the training set, to solve the problem that it is difficult to place sensors on the performer, the author made a two-degree-of-freedom flute-playing robot, which can control the angle and flow rate of the breath, and the sound data of the robot playing the flute is used as the training set. Based on the extensive use of the training set generated by the flute-playing robot, a small human dataset is added. The human-played training set is not precisely quantified, but is artificially divided into several labels (eg, the flow rate is "F" or "P"). 

As mentioned in this paper, some examples have large errors when estimating the angle of breath using this method. The reason may be that there are more parameters to be controlled when people play the flute, and only two parameters, the blowing angle, and the flow rate, are considered in this paper. In addition to the piper mouth shape mentioned in the article, the humidity of the flute-playing robot's air may also affect the sound. Therefore, further research can focus on the acoustic characteristics of the flute, design a flute-playing robot that is more in line with the human flute-playing environment, and obtain the training set with more control parameters.

The production process of the training set in this paper should be clearly described in detail.

Generally deep learning is used for image classification. This paper also converts the sound data into images (Fig. 2, Fig. 6), thereby utilizing the MLP visual deep learning model. In this paper, what are the physical meanings of the x axis, y axis, and the values of each pixel of the image?

For the visual deep learning, the training set contains many labeled images. As mentioned in Section 3.4.3 of this paper, 39,000 data points are recorded, so how many images are there in the training set?

Round 2

Reviewer 1 Report

Dear authors, thanks for submitting your revised version, which looks actually improved. I would have one final major suggestion and few minor ones.

Major: although I appreciate that you added a short paragraph on temperature and humidity of airflow, I would ask you whether you can measure the two parameters of the airflow of the robot and include them in the appropriate section.

Minor follow here below: 

Line 113: ref should be Di Tocco et al. I assume (and in ref list Di Stefano instead of Stefano D. N.,) 

Line 114: it is possible --> The acquired data allowed to study....

Line 116: Linden should be Van der Linden

Line 392: one of the authors, plural

I would also recommend to have the manuscript checked for language to increase readability. 

Reviewer 2 Report

This manuscript can be accepted.

Author Response

Thank you very much for reviewing.

We have revised the manuscript as indicated by reviewer 1.